# Deferoxamine Reduces Inflammation and Osteoclastogenesis in Avulsed Teeth

**DOI:** 10.3390/ijms22158225

**Published:** 2021-07-30

**Authors:** Ko Eun Lee, Seunghan Mo, Hyo-Seol Lee, Mijeong Jeon, Je Seon Song, Hyung-Jun Choi, Hyunsoo Cho, Chung-Min Kang

**Affiliations:** 1Department of Pediatric Dentistry, College of Dentistry, Yonsei University, Seoul 03722, Korea; olivedlr@naver.com (K.E.L.); mosh726@gmail.com (S.M.); mjjeon1107@yuhs.ac (M.J.); songjs@yuhs.ac (J.S.S.); choihj88@yuhs.ac (H.-J.C.); 2Department of Pediatric Dentistry, Kyung Hee University Dental Hospital, Seoul 02447, Korea; snowlee@khu.ac.kr; 3Department of Pediatric Dentistry, Kyung Hee University School of Dentistry, Seoul 02447, Korea; 4Oral Science Research Center, College of Dentistry, Yonsei University, Seoul 03722, Korea; 5Division of Hematology, Department of Internal Medicine, Yonsei University College of Medicine, Seoul 03722, Korea

**Keywords:** deferoxamine, inflammation, infection control, osteoclast(s), bone remodeling/regeneration, bone biology

## Abstract

Replacement and inflammatory resorption are serious complications associated with the delayed replantation of avulsed teeth. In this study, we aimed to assess whether deferoxamine (DFO) can suppress inflammation and osteoclastogenesis in vitro and attenuate inflammation and bone resorption in a replanted rat tooth model. Cell viability and inflammation were evaluated in RAW264.7 cells. Osteoclastogenesis was confirmed by tartrate-resistant acid phosphatase staining, reactive oxygen species (ROS) measurement, and quantitative reverse transcriptase–polymerase chain reaction in teeth exposed to different concentrations of DFO. In vivo, molars of 31 six-week-old male Sprague–Dawley rats were extracted and stored in saline (n = 10) or DFO solution (n = 21) before replantation. Micro-computed tomography (micro-CT) imaging and histological analysis were performed to evaluate inflammation and root and alveolar bone resorption. DFO downregulated the genes related to inflammation and osteoclastogenesis. DFO also reduced ROS production and regulated specific pathways. Furthermore, the results of the micro-CT and histological analyses provided evidence of the decrease in inflammation and hard tissue resorption in the DFO group. Overall, these results suggest that DFO reduces inflammation and osteoclastogenesis in a tooth replantation model, and thus, it has to be further investigated as a root surface treatment option for an avulsed tooth.

## 1. Introduction

Replacement and inflammatory resorption are serious complications that are associated with the replantation of avulsed teeth [1]. Avulsion is associated with damage of the periodontal ligament (PDL), decrease in the viability of PDL cells at the root surface, and the presence of bacterial contamination [2]. Short extra-alveolar time and the use of a suitable temporary storage medium are critical for a good prognosis in replanted teeth [3,4,5].

Sometimes, depending on the site of the trauma and other related conditions, replantation cannot be conducted immediately, and/or an appropriate storage medium cannot be used. Therefore, in cases where the replantation is delayed, there have been several attempts to increase the viability of the tooth by applying root surface treatments with chemical materials [3], such as acidulated fluoride [6], alendronate [7], and tooth enamel protein [8] (Emdogain^®^).

Deferoxamine (DFO) is a Food and Drug Administration–approved iron chelator used to reduce acute and chronic iron overload and aluminum toxicity in patients with chronic kidney disease and patients with thalassemia receiving a large amount of blood transfusions [9]. Recently, the use of DFO has been expanded to regenerative medicine [10], as it can increase angiogenesis in wound healing and bone regeneration and has anti-inflammatory properties [11].

In dentistry, DFO is known to induce osteoblastic and osteogenic differentiation of human periodontal ligament cells [12,13]. As a representative hypoxia-mimicking agent, the angiogenic capacity of DFO was investigated by confirming the increase in vascular endothelial growth factor and upregulation of the related downstream angiogenic pathways. DFO promotes odontoblast differentiation of not only periodontal cells but also dental pulp cells; because of its high healing potential, DFO is also applied to repair dental pulp stem cells [14,15].

Reactive oxygen species (ROS) are essential for the transmission of cell signals and have various physiological functions [16]. It is known that iron chelators, such as DFO, are closely associated with ROS signaling [17]. Excessive ROS can interfere with the balance between the oxidant and antioxidant systems [18]. As shown in various studies on inflammation [19], rheumatoid arthritis [20], and osteoporosis [21], increased ROS levels are also strongly associated with increased bone destruction and inflammation. However, the effects of DFO on ROS production are still controversial. According to the studies of Jiang et al. [22] and Temraz et al. [23], DFO reduces the ROS level in the cells, thereby preventing apoptosis and managing iron homeostasis, whereas other studies concluded that there is increased intracellular ROS generation under DFO [12,14].

Given its safety and potential efficacy, DFO needs to be considered as a candidate for root surface treatments. However, to the best of our knowledge, there has been no study on the effects of DFO on inflammation and osteoclastogenesis in a tooth replantation model. Therefore, in this study, we aimed to assess the effects of DFO on inflammation and osteoclastogenesis in vitro and in replanted rat molar tooth models.

## 2. Results

### 2.1. Cell Viability and Anti-Inflammatory Effect of DFO in RAW264.7 Cells

Cytotoxic properties (indicated by CCK-8) indicated no significant difference in cell viability between the control and dose-dependent DFO treatment groups (Figure 1A). NO release decreased at 20, 50, and 100 μM of DFO compared to the LPS treatment group (*p* < 0.01, Figure 1B). Pro-inflammatory cytokine expression (IL-1β and IL-6) also decreased in the DFO-treated group compared to the LPS-treated group (*p* < 0.05) (Figure 1C,D).

### 2.2. DFO Affects Osteoclast Differentiation in RAW264.7 Cells

To further investigate the role of DFO in osteogenesis, TRAP/ALP staining of RANKL-treated RAW264.7 cells was conducted, resulting in red-stained osteoclasts. Cells in the DFO treatment group showed lighter TRAP staining that those in the RANKL treatment group. Moreover, the higher the DFO concentration, the lower was the TRAP staining intensity (Figure 2). The expression of several genes related to osteoclast differentiation was confirmed by qPCR. TNF receptor associated factor 6 (TRAF6), nuclear factor of activated T-cell, cytoplasmic 1 (NFATc1), TRAP, and cathepsin K (CTSK) expression levels were lower in the DFO treatment group and differed significantly compared to that in the RANKL treatment group (100% of RANKL treatment group, Figure 3B–E). These results suggest that DFO is involved in inhibiting the production of osteoclasts.

Since DFO attenuates osteoclast differentiation, we investigated several well-known factors associated with osteoclastogenesis to determine the signaling pathway. As a result, it was confirmed that treatment with DFO decreased the expression of ROS in RANKL-induced osteoclast differentiation (Figure 3A), and of RelA, known as p65, a subunit of NF-κB [24] (Figure 3F). In addition, as significant regulators of ROS metabolism [25], the expression levels of CREB and PGC-1β were also decreased (Figure 3G,H). On the other hand, the expression of Nrf-2 and the related HO-1, a key signal pathway for antioxidant and anti-inflammatory activity, increased (Figure 3I,J).

### 2.3. Anti-Inflammation and Reduced Bone Loss in Replanted Rat First Molars

In saline, the external root resorption happened heterogeneously and invaded the cementum and dentine irregularly. Extensive absorption was also observed in bones in contact with the replanted tooth (Figure 4B,E). Although inflammatory alveolar bone and root resorption were observed in the DFO group, the thickness and amount of absorbed bone and the amount of resorbed cementum and dentin of the root were small (Figure 4C,F). At the worst score of 3, a wide range of inflammatory resorption in the alveolar bone (average score = 2.40) and extensive resorption of dentin and cementum of the MB root (average score = 2.50) were observed in the saline group. However, in the DFO group, alveolar bone resorption showed a relatively smaller area, at an average score of 1.86. Root resorption was also reduced in most cases, with an average score of 1.95. However, the difference in bone and root resorption between the groups was not statistically significant (Figure 4G,H). The difference in ankylosis was not statistically significant between the saline (average score = 0.30) and DFO (average score = 0.24) groups (Figure 4I). Furthermore, no statistical difference was observed in canal calcification between the saline (mean = 0.40) and DFO (average score = 0.33) groups (Figure 4J). Figure 5 shows the histological analysis of the interstitial space between replanted tooth and bone. The replanted root included fibrous tissue with inflammatory cells, such as lymphocytes, plasma cells, and neutral multinuclear cells. In addition, PDL-like structures were hardly observed in the saline group. An aggressive root replacement resorption was performed not only in the dentine–cementum structure but also in the inner dentine wall of the replanted root (Figure 5B,E). In the DFO group, small area of depression was observed on the root surface, inflammation state was found in the apex 1/3 area, and the middle 1/3 showed relatively good adhesion of the fibers with less expanded PDL space (Figure 5C,F). The differences in the degree of root resorption and inflammation at the epithelial insertion were statistically different (*p* < 0.05) between the two groups (Figure 5H,J). There was no statistical difference in bone resorption and inflammation at the epithelial insertion between the saline (mean = 0.40) and DFO (average score = 0.30) groups (Figure 5G,I), which showed a reducing tendency.

## 3. Discussion

Avulsed teeth that have been replanted are at a risk of infection, inflammation, and root and bone resorption. Although DFO is widely used in various medical conditions [26,27], there have been no studies on the effects of DFO as a root surface treatment option in replanted, avulsed teeth. In the present study, topically applied DFO exerted anti-inflammatory effects and prevented osteoclast differentiation. The in vivo experiment supports the results of the reduced expression of genes related to inflammation and osteoclastogenesis. The Nrf2/HO-1 pathway, one of the mitochondrial biogenesis pathways that reduce NF-κB, ROS, and CREB/PGC-1β signal transduction, might be the mechanism underlying these findings. The schematic signaling pathways of DFO that control inflammation and osteoclastogenesis are represented in Figure 6.

DFO is an effective iron chelator, which binds to iron at a one-to-one ratio [28]. Iron is important for the survival of cells and metabolism at the cellular and systemic states [23]. Complex mechanisms tightly regulate iron absorption and release to maintain iron homeostasis. Disruption of iron homeostasis leads to a variety of disease states, including neurologic diseases, cardiovascular complications, infection, and cancer [29]. In particular, DFO is widely used in medical patients with iron overload, such as those with thalassemia and sickle cell anemia, and in acute iron poisoning in small children. However, in recent years, it not only functions as an iron chelator but is also gaining more attention because of its antioxidant capability. This is an extension of iron chelation, is involved in iron homeostasis, and is closely related to the concentration of ROS, an important factor in cell signaling [11].

We found that DFO application can reduce inflammation, consistent with previous studies [30]. Inflammation in the replanted tooth is closely related to rapid destruction of the root surface and alveolar bone loss, which can lead to the loss of a tooth [31]. The pro-inflammatory cytokine, IL-1, stimulates bone resorption and interferes with bone production [32]. It stimulates the growth and promotes the differentiation and maturation of osteoclast progenitor cells [33]. The expression of IL-1 and IL-6 is evidence that inflammatory conditions can lead to bone resorption [34]. This reaction can rapidly degrade the cementum and dentin of the tooth [35]. NO is also connected with the pathogenesis of osteoporosis and induces bone loss [36].

In vitro studies have revealed through TRAP staining that DFO inhibits osteoclastogenic differentiation, and the activity of osteoclast-related factors (TRAF6, NFATc1, TRAP, and CTSK) has also observed. The RANK/RANKL-dependent signaling pathway regulates osteoclastogenesis [37]. TRAF6 is a key protein that transmits the RANKL signals to the nucleus through pathways such as NF-κB and MAPK. The expression of NFATc1 plays an important role in the activation of osteoclasts via RANK/RANKL-dependent signaling. NFATc1, in turn, increases the expression of TRAP and CTSK, which are necessary for releasing acid and producing the degrading enzymes that lead to bone resorption [38]. DFO inhibits the expression of factors related to the differentiation of osteoclasts in bone [39].

The high labial intracellular iron pool can induce ROS generation [40]. Moreover, ROS and free radicals are essential for enhancing osteoclast differentiation and inflammation. Excessive ROS can result in oxidative stress and interrupt cellular homeostasis, especially under normal physiological conditions. ROS produced by osteoclasts stimulate and facilitate the resorption of bone tissue [41]. Our results confirmed that DFO can reduce ROS levels and work as an antioxidant [11].

The activity of NFATc1, a major transcription factor for osteoclast differentiation, can also be reduced by DFO, because of biogenesis-related genes. PGC-1β is a member of the transcription coactivators that regulate cellular energy metabolism, stimulate mitochondrial biogenesis, and promote remodeling [42]. In addition, CREB binds to cAMP response elements (CREs) in the DNA sequence and regulates the transcription of the genes [43]. This pathway activates osteoclastogenesis, but if the ROS levels decrease after DFO treatment, its function is also degraded.

DFO not only reduces ROS but also induces the activity of Nrf2, which was also confirmed in our study. Nrf2 is an essential transcription factor that regulates antioxidant defense gene expression, such as that of *HO-1*. Recently, antioxidant, anti-inflammatory, and immunomodulatory effects of HO-1 have been observed in vascular cells [44], and they have a substantial effect on osteoclastogenesis [45]. The Nrf2/HO-1pathway reduces intracellular ROS levels and inhibits NF-κB signaling [46], which was also confirmed through our experiments.

Here, the results of the in vivo study confirmed that DFO reduces inflammation and osteoclastogenesis in 28 days compared to that in the saline group. The DFO-treated group showed reduced pulp damage, such as obliteration, and less ankylosis or root resorption with damaged PDL. Müller et al. investigated the complications of avulsed teeth and concluded that 33% of teeth were extracted during the mean observation period of 3.5 years specifically because of inflammatory resorption and reported that the unpredictable and poor results could be reduced by regenerating the PDL of the avulsed tooth [31].

To the best of our knowledge, this is the first in vivo study to investigate the potential of DFO as a root surface treatment in tooth replantation. DFO is a widely used iron chelator, whose safety is confirmed by the Food and Drug Administration of the United States; it has advantages over other treatment options (such as acidulated fluoride, alendronate, and tooth enamel protein) in the replantation of teeth, in terms of cost and availability. However, the treatment dosage of DFO for optimal efficacy and the sufficient safety margin have not been assessed in tooth replantation, and these warrant further investigation.

However, this study had some limitations. First, bone remodeling, including bone resorption, is a result of the crosstalk between osteoblasts and osteoclasts, but we only focused on osteoclasts. In addition, although the representative NF-κB pathway was examined in terms of the osteoclastogenesis mechanism, pathways involving MAPK or Akt were not investigated despite their importance in osteoclast differentiation. Second, as discussed in the results, ankylosis and canal obliteration are among the long-term adverse effects of tooth trauma and can be more accurately identified when performing a long-term follow-up.

## 4. Materials and Methods

### 4.1. Cell Culture

RAW264.7 cells were obtained from the Korean Cell Line Bank and cultured in Dulbecco’s modified Eagle medium (DMEM; Invitrogen, Carlsbad, CA, USA), containing 10% fetal bovine serum (FBS; Invitrogen), 100 U/mL penicillin, 100 µg/mL streptomycin (Invitrogen), and 0.2% amphotericin B (Invitrogen), at 37 °C in 5% CO_2_.

### 4.2. Cell Viability Assay

Measurement of cell viability was performed using Cell Counting Kit-8 (CCK-8) (Dojindo Laboratories, Kumamoto, Japan). RAW264.7 cells were plated onto a 24-well plate (25,000 cells/well) and incubated for 24 h, followed by treatment with different doses of DFO (Sigma, St. Louis, MO, USA) and incubation for another 24 h. Subsequently, the quantity of water-soluble colored formazan from the CCK-8 assay, formed by the activity of dehydrogenases in living cells, was measured using a spectrophotometer (Benchmark Plus microplate spectrophotometer, Bio-Rad Laboratories Inc., Hercules, CA, USA) at 450 nm. All experimental data were obtained from three independent experiments, with each sample run in triplicate.

### 4.3. Nitric Oxide Assay

RAW264.7 cells were treated with DFO (0, 10, 20, 50, and 100 µM) for 2 h, and lipopolysaccharide (LPS) (0.5 µg/mL; Sigma) was added. Cells were incubated for 20 h at 37 °C in CO_2_. To determine the nitrite release in the culture media, presumed to reflect the nitric oxide (NO) levels, Griess reaction was used, wherein 100 µL cell culture medium was mixed with 100 µL of Griess reagent (Invitrogen) and incubated at room temperature for 30 min. The NO concentration was determined at 540 nm using a spectrophotometer (Bio-Rad Laboratories Inc.).

### 4.4. In Vitro Osteoclastogenesis Assays

RAW264.7 cells were plated in 24-well plates (1 × 10^4^ cells/well) and cultured in DMEM treated with DFO (0, 10, 20, 50, and 100 µM) for 2 h. Next, DMEM was changed with Minimum Essential Medium (MEM) Alpha (Invitrogen) containing 10% FBS and 50 ng/mL of receptor activator of nuclear factor-κB ligand (RANKL) (PeproTech Inc., Rocky Hill, NJ, USA), for 3 days at 37 °C in a CO_2_ incubator.

### 4.5. Tartrate-Resistant Acid Phosphatase (TRAP) Staining

RAW264.7 cells were plated in 24-well plates (1 × 10^4^ cells/well) and cultured in DMEM treated with DFO (0, 10, 20, 50, and 100 µM) for 2 h. Next, DMEM was replaced with Minimum Essential Medium Alpha (Invitrogen), containing 10% FBS and 50 ng/mL of receptor activator of nuclear factor-κB ligand (RANKL) (PeproTech Inc., Rocky Hill, NJ, USA), and the cells were incubated for 3 days at 37 °C in a CO_2_ incubator. Finally, osteoclast formation was confirmed using a tartrate-resistant acid phosphatase (TRAP)/alkaline phosphatase staining kit (WAKO, Osaka, Japan).

### 4.6. ROS Assessment

ROS levels were assessed by incubating cells with H_2_DCFDA (20 µM; Abcam, Cambridge, MA, USA) for 30 min at 37 °C. After incubation with H_2_DCFDA, cells were washed with PBS and assessed for fluorescence intensity by employing a BD LSR II flow cytometer (BD Biosciences, San Jose, CA, USA). Data were analyzed using FCS express Flow Cytometry Software (De Novo, Glendale, CA, USA).

### 4.7. Quantitative Real-Time Polymerase Chain Reaction

The integrity and concentration of the total RNA, isolated using an RNeasy Mini kit (Qiagen, Valencia, CA, USA), were evaluated using a spectrophotometer (NanoDrop ND-2000, Thermo Scientific, Waltham, MA, USA). Next, 500 ng RNA aliquots were reverse transcribed to cDNA using a Maxime RT premix kit (oligo d(T)15 primer; Intron Biotechnology, Seongnam, Gyeonggi, Korea). A quantitative real-time polymerase chain reaction (qPCR) assay was performed with SYBR Premix Ex Taq (Takara Bio Inc., Otsu, Japan) and a real-time PCR system (ABI 7300, Applied Biosystems, Carlsbad, CA, USA), as per manufacturer’s instructions. The qPCR conditions were 95 °C for 10 s, followed by 40 cycles of 95 °C for 5 s and 60 °C for 30 s, with a final 5 min extension at 72 °C. The annealing procedures for all primers were performed at 60 °C. Expression of each gene was normalized to that of beta-actin, and the relative expression levels of the target genes were calculated using the 2^−ΔΔCt^ method. The oligo-nucleotide primers are listed in Table 1.

### 4.8. Rat Models

#### 4.8.1. Replantation Procedure in Rat Maxillary First Molar

The maxillary first molar of 31 six-week-old male Sprague–Dawley rats (200–250 g) (Orient Bio, Seoul, Korea) was used for tooth replantation, in accordance with the ethical guidelines and regulations approved by the Institutional Animal Care and Use Committee of Yonsei University (#2018-0214). DFO, diluted to 500 µM in saline, was used for the in vivo study. To facilitate the extraction, 0.4% β-aminopropionitrile (Sigma, St. Louis, MO, USA) mixed in distilled water was administered to the rats along with the feed 3 days prior to extraction. The maxillary left right molar was atraumatically extracted under anesthesia with ketamine (0.1 mL/100 g, Yuhan, Dongjak, Seoul, Korea) and xylazine (0.05 mL/100 g, Bayer Korea, Dongjak, Seoul, Korea). After performing peritomy using explorer, the tooth was extracted with minimal trauma from the alveolar socket using tissue forceps. To determine the drug effect on PDL healing, the extracted teeth were randomly divided into two groups based on the treatment. In the control group (n = 10), the extracted teeth were stored in 50 mL of 0.9% physiological saline (pH 6.0) at 4 °C for 5 min and cotton soaked with 0.9% physiological saline was applied to the alveolar socket to induce thrombus removal and hemostasis. In the experimental group (n = 21), the extracted teeth were stored in 500 µM DFO solution (pH 7.22) at 4 °C for 5 min, and cotton soaked with 500 µM DFO solution was applied to the alveolar socket. Immediately after root surface treatment, the teeth were replanted in the original sockets. After replantation, all animals received a single intramuscular dose of 20,000 UI penicillin G benzathine (Tokyo Chemical Industry Co., Ltd., Tokyo, Japan). After four weeks of replantation, the animals were euthanized under ketamine anesthesia. The maxilla was acquired and fixed in 10% formalin after washing with saline. For each animal, four different investigators were involved as follows: a first investigator took care of animals and was responsible for the anesthesia, whereas a second investigator was responsible for the tooth extraction. A third investigator, the only person aware of the treatment group, administered the treatment materials to the divided group. The second investigator replanted the extracted teeth and gave antibiotics. Finally, a fourth investigator assessed micro-CT images and histological analysis scores.

#### 4.8.2. Micro-Computed Tomography Image Analysis and Histological Analysis

The extracted maxilla was scanned at 10 µm intervals using a micro-computed tomography (micro-CT) system (Quantum FX micro-CT, Perkin Elmer, Norwalk, CT, USA), and the tooth images were reconstructed using a software (TRI-3D, Ratoc System Engineering Co., Ltd., Tokyo, Japan). The mesiobuccal (MB) root was chosen as it is the largest and allows clear observation of changes. A longitudinal image of the replanted tooth was obtained through the long axis of the MB root in the mesiodistal direction. The scoring standards are listed in Table 2. The extension of the bone surface resorption area in contact with the root was measured in micrometers. Then, the ratio of the total surface of the bone in contact with the root was calculated.

#### 4.8.3. Histological Analysis

After the micro-CT scan, the maxilla was decalcified with 10% ethylenediaminetetraacetic acid (pH 7.4; Fisher Scientific Co., Houston, TX, USA) for three weeks at room temperature. The degree of decalcification was confirmed by radiographical analysis. After cryotomy, a 20–25 μm thick section of the maxillary first molar was cut from the sagittal plane to the apical axis, and tissues that were well connected with the apical and apical roots were selected, stained with hematoxylin and eosin, and observed using an optical microscope (Axio Lab, Zeiss, Germany). The degree of inflammatory resorption of the roots, substitutional substitution, PDL status, and degree of inflammation were observed for each group using an optical microscope.

For quantitative analysis, similar to the micro-CT image analysis, only the MB root was observed. In the longitudinal section, the characteristics listed in Table 3 were investigated. The criteria defined in the study for scoring in micro-CT image analysis and histological examination were in accordance with those described by Poi et al. [31,47,48].

### 4.9. Statistical Analyses

All in vitro experiments were performed, at least, in triplicate. All statistical analyses were performed with SPSS (version 25.0, SPSS Inc., Chicago, IL, USA). The normality of the in vitro and in vivo data was evaluated using the Shapiro–Wilk test (*p* < 0.05). Mann–Whitney U test (*p* < 0.05) was used to compare the data of control and experimental, LPS treatment and experimental, RANKL treatment and experimental, and saline and DFO groups.

## 5. Conclusions

Our study showed that DFO can prevent inflammation and bone resorption and that it is a treatment candidate that can be applied to the root surface and alveolar bone before replantation. To the best of our knowledge, this is the first in vivo study in which DFO has been applied to teeth. The results of our study serve as a basis for developing a novel root treatment for tooth replantation using DFO.

## Figures and Tables

**Figure 1 ijms-22-08225-f001:**
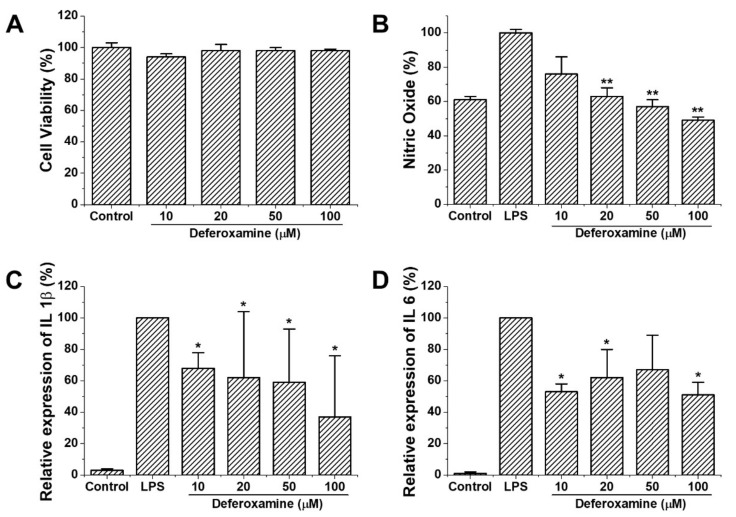
Cell viability and anti-inflammatory effect of deferoxamine in RAW264.7 cells. (**A**) The cell viability of DFO was assessed using a CCK-8 assay kit in RAW264.7 cells. (**B**) NO release of DFO in RAW264.7 cells. Cells were incubated in the presence of different concentrations of DFO and 0.1 µg/mL LPS for 20 h. Then, the culture supernatant was analyzed for NO (**C**, **D**) IL-1β and IL-6 expression at different concentrations DFO-treated RAW 264.7 cells analyzed with qPCR. (**A**–**D**) Cells were treated with different concentrations of DFO. The data are expressed as mean ± standard deviation. Data show mean ± standard deviation values of three independent experiments. * *p* < 0.05 and ** *p* < 0.01 indicate significant differences compared to the control value (100%).

**Figure 2 ijms-22-08225-f002:**
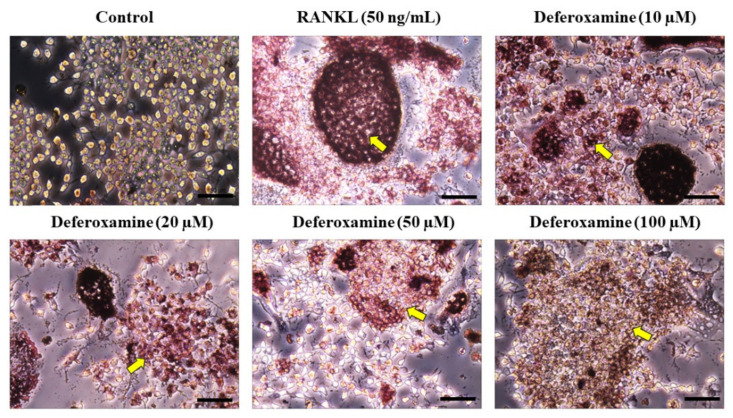
Osteoclast differentiation of DFO in RAW264.7 cells. The cell viability of DFO-treated RAW 264.7 cells was assessed using the CCK-8 assay kit. Tartrate-resistant acid phosphatase (TRAP) staining of DFO in RAW264.7 cells. Cells were applied with different concentrations of DFO for 2 h followed by treatment with 50 ng/mL receptor activator of nuclear factor-κB ligand (RANKL) for 3 days. Subsequently, RAW264.7 cells were fixed and stained to detect TRAP. The osteoclasts were stained red (yellow arrow). Scale bars = 50 µm.

**Figure 3 ijms-22-08225-f003:**
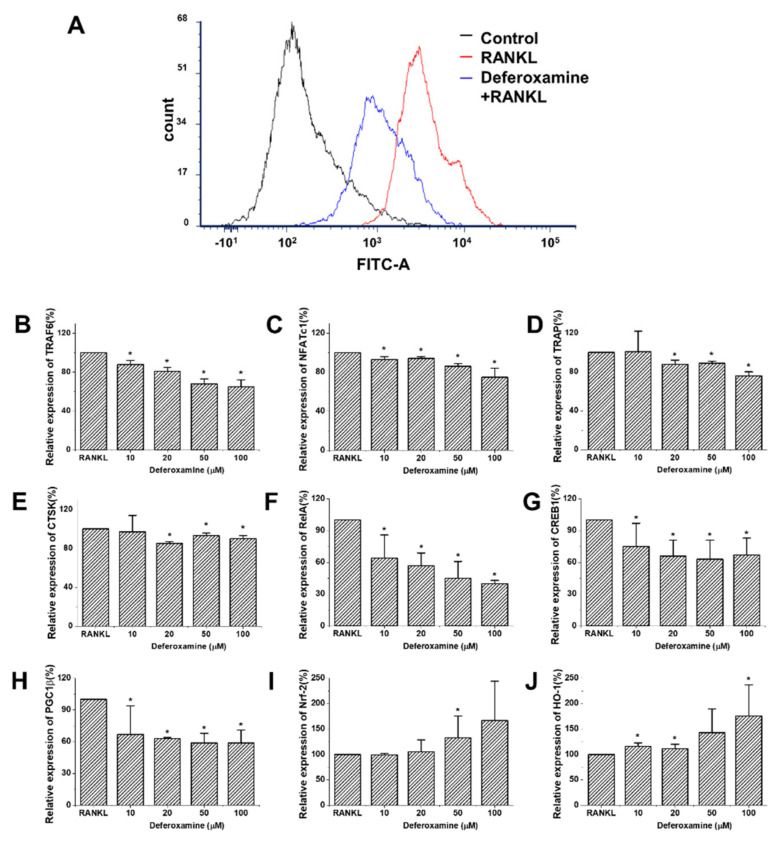
Intracellular reactive oxygen species and related genes with osteoclast differentiation treated with DFO in RAW264.7 cells. (**A**) Intracellular ROS levels determined using an H_2_DCFDA ROS probe and analyzed with FCS express flow cytometry. (**B**–**J**) Changes in the expression of the osteoclast-related genes of DFO in RAW264.7 cells. Cells were pretreated with different concentrations of DFO for 2 h followed by treatment with 50 ng/mL RANKL for 3 days. RNA was isolated from RAW264.7 cells, and cDNA was synthesized. Expression levels of tartrate-resistant acid phosphatase (TRAP), TNF receptor associated factor 6 (TRAF6), nuclear factor of activated T cell, cytoplasmic 1 (NFATc1), and cathepsin K (CTSK) were evaluated using quantitative RT-PCR relative to the RANKL treatment group (normalized to 100%). Expression levels of v-rel avian reticuloendotheliosis viral oncogene homolog A (RelA), cyclic AMP-responsive element-binding protein 1 (CREB1), peroxisome proliferator-activated receptor gamma coactivator-1beta (PGC-1β), heme oxygenase-1 (HO-1), and nuclear factor erythroid 2–related factor 2 (Nrf-2) were evaluated using quantitative RT-PCR relative to the RANKL treatment group (normalized to 100%). Data were obtained from five separate experiments, with all samples run in duplicate. The data are expressed as mean ± standard deviation values. The expression of the genes differed significantly; t-test and Mann–Whitney U test, * *p* < 0.05.

**Figure 4 ijms-22-08225-f004:**
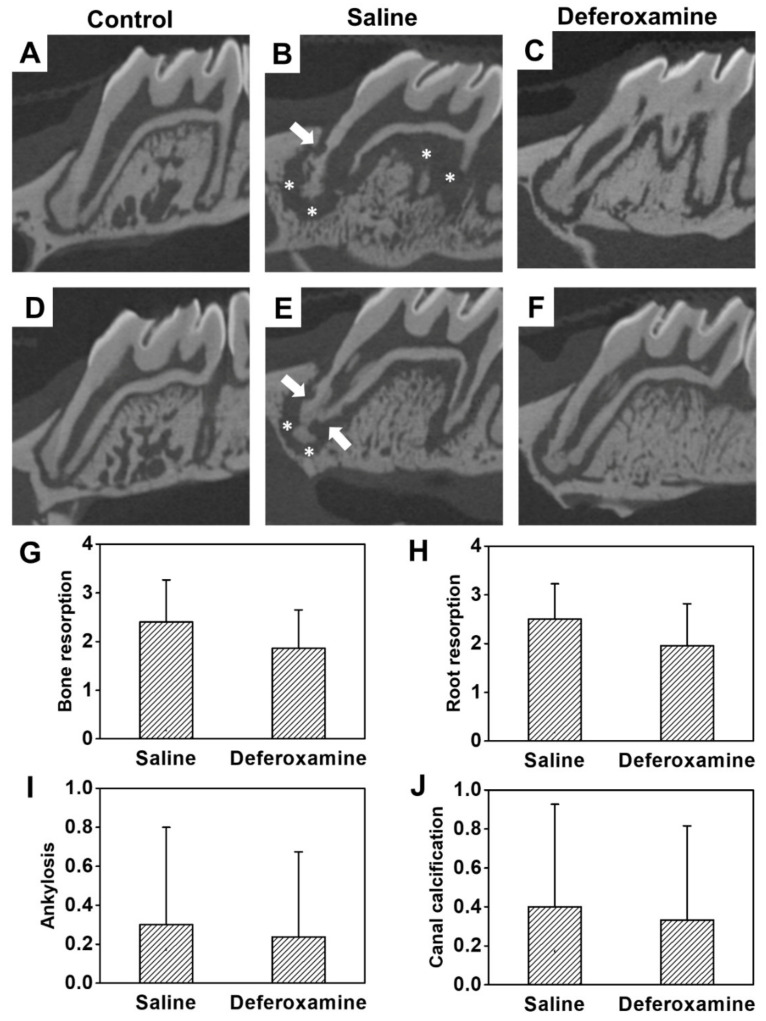
Micro-computed tomography images of rat maxillary first molars 4 weeks after replantation. (**A**,**D**) Control groups: The teeth without extraction and replantation. Normal root surface and alveolar socket. (**B**,**E**) Saline groups: Before replantation, 0.9% physiological saline was applied to the extracted teeth surface and alveolar socket. Severe root resorption (arrows) and extensive alveolar bone resorption (asterisks) were observed. (**C**,**F**) DFO groups: 500 µM DFO solution was applied to the surface of the extracted teeth and alveolar socket. A relatively short range of bone resorption (asterisk) was observed compared to that in the saline groups. (**G**–**J**) Evaluated scores, represented as mean ± standard deviation, for (**G**) bone resorption, (**H**) root resorption, (**I**) ankylosis, and (**J**) canal calcification in the saline and DFO groups. However, there was no statistically significant difference between the saline and DFO groups.

**Figure 5 ijms-22-08225-f005:**
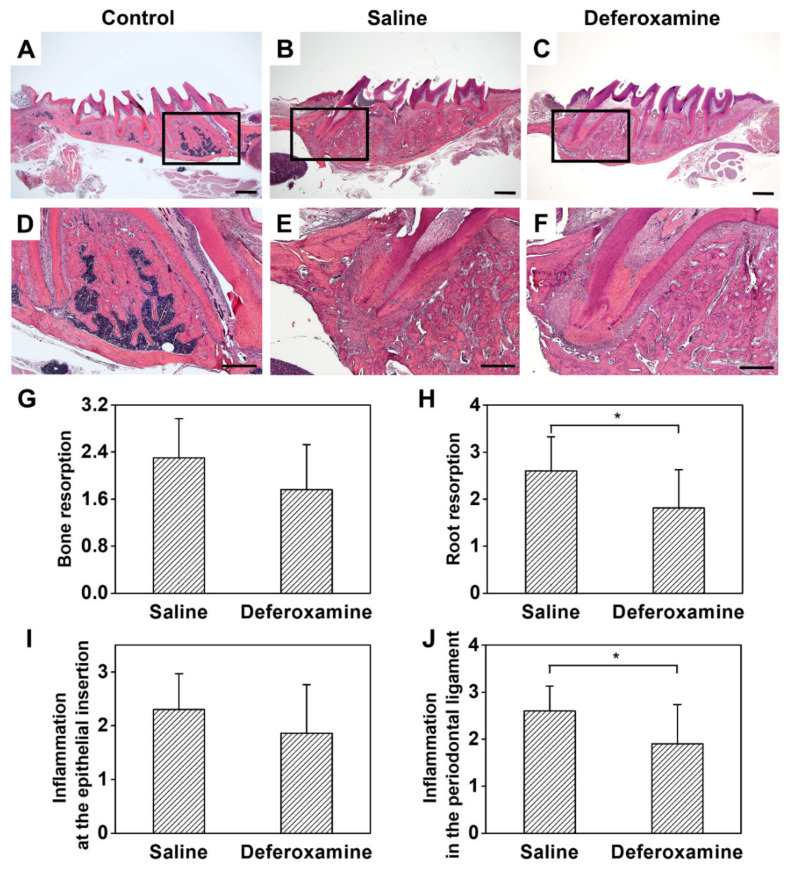
Histological analysis of rat maxillary first molars 4 weeks after replantation. (**A**,**D**) Control groups: The teeth without extraction and replantation. Normal root surface and alveolar socket. (**B**,**E**) Saline groups: Many cells related with inflammation were observed in the periodontal ligament space, severe inflammatory root resorption was found on all surfaces of the root. (**C**,**F**) DFO groups: Localized root resorption in the apical third happened. (**G**–**J**) Evaluated scored, expressed as mean ± standard deviation, for (**G**) bone resorption, (**H**) root resorption, (**I**) inflammation at the epithelial insertion, and (**J**) inflammation in the periodontal ligament in the saline and DFO groups. Hematoxylin–eosin staining, Mann–Whitney U test, * *p* < 0.05 (applies to **G**–**J**). Scale bars: (**A**–**C**): 1 mm, (**D**–**F**): 500 µm.

**Figure 6 ijms-22-08225-f006:**
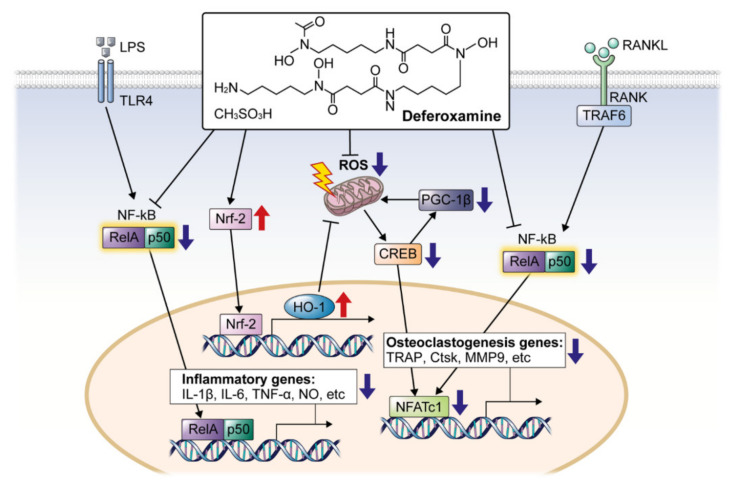
Scheme of a working model for inhibition of inflammation and osteoclast differentiation by DFO in RAW264.7 cells.

**Table 1 ijms-22-08225-t001:** Primers used for quantitative real time polymerase chain reaction analysis.

Gene	Forward Primer Sequence (5′–3′)	Reverse Primer Sequence (5′–3′)
*CTSK*	GGGATGTTGGCGATGCA	CCAGCTACTTGAGGTCCATCTTC
*CREB1*	TGTACCACCGGTATCCATGC	TGGATAACGCCATGGACCTG
*GAPDH*	CTGGCACAGGGTATACAGGGTTAG	ACTGGTGCCGTTTATGCCTTG
*HIF1a*	AGCTTGCTCATCAGTTGCCA	CCAGAAGTTTCCTCACACGC
*HO-1*	GAAATCATCCCTTGCACGCC	CCTGAGAGGTCACCCAGGTA
*IL 1β*	GTTCCCCAACTGGTACATCA	CCATACTTTAGGAAGACACGG
*IL 6*	GTTGCCTTCTTGGGACTGATG	ATCAGAATTGCCATTGCACAA
*NFATc1*	CACTGGCGCTGCAACAAGA	CATTCCGGAGCTCAGCAGAATAA
*Nrf-2*	TGAAGCTCAGCTCGCATTGA	TGCTCCAGCTCGACAATGTT
*PGC-1β*	CTCTGACACGCAGGGTGG	AGTCAAAGTCACTGGCGTCC
*RelA*	CTCCTGAACCAGGGTGTGTC	GAGAGACCATTGGGAAGCCC
*TRAF6*	CTACCCGCTTTGACATGGGT	CACCTCTCCCACTGCTTGTT
*TRAP*	CAAAGGTGCAGCCTTTGTGTC	TCACAGTCCGGATTGAGCTCA
*VEGF*	TCCGAAACCATGAACTTTCTGC	AGCTTCGCTGGTAGACATCC
*ACTB*	TCACCATGGATGATGATATCGC	GGAATCCTTCTGACCCATGC

**Abbreviations:** CTSK, gene encoding cathepsin K; CREB1, gene encoding CAMP responsive element binding protein 1; GAPDH, gene encoding glyceraldehyde-3-phosphate dehydrogenase; HIF1a, hypoxia-inducible factor 1 alpha; HO-1, heme oxygenase-1; IL 1β, interleukin 1 beta; IL 6, interleukin 6; NFATc1, gene encoding nuclear factor of activated T-cells, cytoplasmic 1; Nrf-2, nuclear factor erythroid 2–related factor 2; PGC-1β, gene encoding PPARG coactivator 1 beta; RelA, gene encoding REL-associated protein; TRAF6, TNF-receptor-associated factor 6; TRAP, gene encoding tartrate-resistant acid phosphatase; VEGF, vascular endothelial growth factor; and ACTB, beta-actin.

**Table 2 ijms-22-08225-t002:** Scores for quantitative analysis in micro-computed tomography image.

Characterization	Score
A. Bone resorption	
No resorption present	0
Resorption occurred in <1/3 of the bone surface	1
Resorption occurred in >1/3 and <2/3 of the bone surface	2
Resorption occurred in >2/3 of the bone surface	3
B. Root resorption	
No resorption present	0
Resorption occurred in <1/3 of the bone surface	1
Resorption occurred in >1/3 and <2/3 of the bone surface	2
Resorption occurred in >2/3 of the bone surface	3
C. Ankylosis (direct fusion of bone to root)	
Absence	0
Presence	1
D. Canal calcification	
Absence	0
Presence	1

**Table 3 ijms-22-08225-t003:** Scores for quantitative analysis in histological examination.

Characterization	Score
A. Bone resorption	
No resorption present	0
Resorption occurred in <1/3 of the bone surface	1
Resorption occurred in >1/3 and <2/3 of the bone surface	2
Resorption occurred in >2/3 of the bone surface	3
B. Root resorption	
No resorption present	0
Resorption occurred in <1/3 of the bone surface	1
Resorption occurred in >1/3 and <2/3 of the bone surface	2
Resorption occurred in >2/3 of the bone surface	3
C. Inflammation at the epithelial insertion	
Absence or occasional presence of inflammatory cells	0
Inflammatory process restricted to the lamina propria of the internal part of the epithelium	1
Inflammatory process extending apically to the small portion of the connective tissue underlying the lamina propria of the internal portion of the gingival epithelium	2
Inflammatory process reaching the proximity of the alveolar bone crest	3
D. Inflammation in the periodontal ligament	
Absence or occasional presence of inflammatory cells	0
Inflammatory process presents only in the apical, coronal, or small lateral area of the PDL	1
Inflammatory process reaching more than half of the lateral PDL of the root	2
Inflammatory process in the whole PDL	3

## Data Availability

Data are contained within the article.

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
