# Peer review of "Deferoxamine Reduces Inflammation and Osteoclastogenesis in Avulsed Teeth"

_ijms, 2021, doi:10.3390/ijms22158225_

Round 1
Reviewer 1 Report
The aim of this study was to test the effect of deferoxamine (DFO) on inflammatory response of replanted of avulsed teeth. Cell and animal experiments were carried out in this study. Over all this manuscript is well in writing. However the reviewer has the following comments and suggests may help the author to improve their manuscript.
- Line 66, the effects of DFO on ROS production “are still controversial”. Please provide references to show both positive and negative reports.
- Line 109, “(A) Cell viability of DFO was assessed using CCK-8 assay kit in RAW264.7 cells.” This sentence should be a typo.
- Figure 2, for easy to read, please arrows in the figure to identify the stained markers.
- According to the result showed in Line 137-138, difference in bone and root resorption between the saline and DFO groups was not statistically significant. I suggest the author to revise their title to be “…..Deferoxamine reduces inflammation and osteoclastogenesis related factors in avulsed tooth” and remove the “in vitro and in a rat model” part.
- Line 142, Figure 6 should be Figure 5.
- Line 143-144, “the replanted root included fibrous tissue with inflammatory cells, such as lymphocytes, plasma cells, and neutral multinuclear cells.” Please use arrows and/or markers to identify these cells as well as the so-called PDL-like structures, respectively.
- DFO is an effective iron chelator. The iron was not tested in this study. However a paragraph can be added to discuss this issue.
- What is the purpose of cell viability test on Raw264.7? Why the author did not test the cell viability effect of DFO on bone cells or PDL cells?
- Line 289-293, (1X104 cells/well) and CO2 should be revised.
Author Response
I would like to express my appreciation for your advice.
We agreed with your comments and introduced corrections where required in the manuscript.
I look forward to your response and hope that the revised manuscript is suitable for publication in your journal.
Yours sincerely,
Chung-Min Kang

Reviewer 2 Report
The authors were aimed to assess the effects of Deferoxamine (DFO) on inflammation and osteoclastogenesis in vitro, and in replanted rat molar tooth models.
The study covers some issues that have been overlooked in other similar topics. The study was conducted with a good scientifically sound. Moreover, the study is easy to follow, but some issues should be improved before publication.
The manuscript needs moderate English change and grammar correction. Please also check typos thorough the text.
Conclusion paragraph required a general revision to eliminate redundant sentences and to add some "take-home message".
Author Response

(The authors gave the same response as above.)
